# Synergistic Antibacterial Efficacy of Melittin in Combination with Oxacillin against Methicillin-Resistant *Staphylococcus aureus* (MRSA)

**DOI:** 10.3390/microorganisms11122868

**Published:** 2023-11-27

**Authors:** Ana Flávia Marques Pereira, Alessandra Aguirra Sani, Tatiane Baptista Zapata, Débora Silva Marques de Sousa, Bruno César Rossini, Lucilene Delazari dos Santos, Vera Lúcia Mores Rall, Carla dos Santos Riccardi, Ary Fernandes Júnior

**Affiliations:** 1The Center for the Study of Venoms and Venomous Animals of UNESP (CEVAP), São Paulo State University (UNESP), Botucatu 18619-002, São Paulo, Brazil; ana.f.pereira@unesp.br; 2Department of Chemical and Biological Sciences, Microbiology and Immunology Sector, Institute of Biosciences of Botucatu (IBB), São Paulo State University (UNESP), Botucatu 18618-689, São Paulo, Brazil; alessandra.sani@unesp.br (A.A.S.); tatiane.zapata@unesp.br (T.B.Z.); debora.sousa@unesp.br (D.S.M.d.S.); vera.rall@unesp.br (V.L.M.R.); 3Institute of Biotechnology (IBTEC), São Paulo State University (UNESP), Botucatu 18607-440, São Paulo, Brazil; bruno.rossini@unesp.br (B.C.R.); lucilene.delazari@unesp.br (L.D.d.S.); 4Graduate Program in Tropical Diseases and Graduate Program in Research and Development (Medical Biotechnology), Botucatu Medical School (FMB), São Paulo State University (UNESP), Botucatu 18618-687, São Paulo, Brazil; 5Department of Bioprocesses and Biotechnology, Faculty of Agricultural Sciences (FCA), São Paulo State University (UNESP), Botucatu 18610-034, São Paulo, Brazil; carla.riccardi@unesp.br

**Keywords:** antibiofilm activity, antibiotic resistance, antimicrobial peptides, biofilm, proteomics, synergism

## Abstract

Methicillin-resistant *Staphylococcus aureus* (MRSA) often cause infections with high mortality rates. Antimicrobial peptides are a source of molecules for developing antimicrobials; one such peptide is melittin, a fraction from the venom of the *Apis mellifera* bee. This study aimed to evaluate the antibacterial and antibiofilm activities of melittin and its association with oxacillin (mel+oxa) against MRSA isolates, and to investigate the mechanisms of action of the treatments on MRSA. Minimum inhibitory concentrations (MICs) were determined, and synergistic effects of melittin with oxacillin and cephalothin were assessed. Antibiofilm and cytotoxic activities, as well as their impact on the cell membrane, were evaluated for melittin, oxacillin, and mel+oxa. Proteomics evaluated the effects of the treatments on MRSA. Melittin mean MICs for MRSA was 4.7 μg/mL and 12 μg/mL for oxacillin. Mel+oxa exhibited synergistic effects, reducing biofilm formation, and causing leakage of proteins, nucleic acids, potassium, and phosphate ions, indicating action on cell membrane. Melittin and mel+oxa, at MIC values, did not induce hemolysis and apoptosis in HaCaT cells. The treatments resulted in differential expression of proteins associated with protein synthesis and energy metabolism. Mel+oxa demonstrated antibacterial activity against MRSA, suggesting a potential as a candidate for the development of new antibacterial agents against MRSA.

## 1. Introduction

Bacterial resistance has emerged as a critical global concern, resulting in high mortality rates and soaring healthcare costs with prolonged periods of hospitalization, further aggravated by the SARS-CoV-2 pandemic, where antibiotics were used incorrectly, and this alarming rise of antibiotic resistance coincides with a decline in antibiotic development efforts [1]. In 2019, about five million deaths worldwide were attributed to infections caused by resistant bacteria [2].

Initially restricted to hospital settings, antimicrobial resistance began to emerge in the mid-1990s as community isolates, subsequently spreading worldwide. One notable example is methicillin-resistant *Staphylococcus aureus* (MRSA), which may lead to significant difficult treatment infections with high mortality, including chronic wound infections, endocarditis, bloodstream infections, osteomyelitis, pneumonia, bacteremia, and device-related infections [3,4] often associated with biofilm production; about 80% of nosocomial infections are biofilm-mediated [5]. Biofilm, characterized by their resilience, are notoriously challenging to eradicate, and can exhibit antibiotic resistance levels ranging from 100 to 1000 times higher than planktonic bacteria with persistent cells that can disperse [3,4]. The anti-biofilm compounds have two alternatives to combat the biofilms, acting in biofilm formation or destroying mature biofilms [5]. 

Antimicrobial peptides (AMPs) are naturally occurring defense molecules found in various life forms, including insects, plants, microorganisms, and mammals. These peptides commonly exhibit amphipathic and cationic properties, with antibacterial, antifungal, antiviral, and antiparasitary activities [6]. Melittin, a prominent AMP, constitutes approximately 50% of the dry weight of *Apis mellifera* bee venom, comprising 26 amino acid residues, exhibits a broad spectrum of action, and is less likely to induce resistance [6,7]. Melittin already showed antibacterial and anti-biofilm activities against uropathogenic *Escherichia coli* [8]. 

Natural products, such as AMPs, can be associated with antibacterial drugs to enhance antibacterial effectiveness, reduce antimicrobial doses, mitigate adverse effects, prevent the development of resistance, and act through diverse mechanisms of action [4]. Cucurbitacin B, an active component from *Ecballium elaterium* L. plant, has demonstrated in vitro synergistic effects when combined with oxacillin, an β-lactam antibiotic that acts binding on penicillin-binding protein (PBP), against *S. aureus* isolates [9]. Melittin, when associated with ciprofloxacin, gentamicin, rifampin, and vancomycin, has demonstrated in vitro antibacterial and anti-biofilm activities against multidrug-resistant pathogens [10]. Another area of investigation involves the conjugation of beta-lactam antibiotics or their hybrids with AMPs to combat multidrug-resistant and gram-negative bacteria in vitro [11,12].

Thus, the aims of this study were to investigate the antibacterial activity of melittin against MRSA isolates, and to evaluate the antibiofilm properties of melittin, oxacillin, and their combination (mel+oxa), along with elucidating their potential mechanisms of action against MRSA. This research highlights the promise of melittin in combination with oxacillin as a prospective candidate for the development of novel antibacterial strategies against MRSA.

## 2. Materials and Methods

### 2.1. Bacterial Isolates and Chemicals

The MRSA isolates included ATCC 33591 and one MRSA isolate obtained and stored from The Clinical Hospital from Botucatu Medical School, São Paulo State University (UNESP). Melittin, from *Apis mellifera* bee venom, and oxacillin were purchased from Sigma-Aldrich (Merck™, Darmstadt, Germany) with purities of 815–950 μg/mg and ≥65%, respectively. Cephalothin was provided by Teuto^®^ (Anápolis, Brazil).

### 2.2. Minimum Inhibitory Concentration (MIC) and Minimum Bactericidal Concentration (MBC)

For MIC determination, 100 microliters of the test products were dispensed into a 96-well plate, covering a concentration range from 256 to 0.5 μg/mL, using Mueller Hinton broth (MH). The plate was then inoculated with 100 μL of MRSA isolates at a concentration of 10^5^ colony-forming units per mL (CFU/mL). Subsequently, the plates were incubated at 37 °C for 24 h. MBC values were determined by subculturing the contents of the incubated plates onto MH agar plates [13].

### 2.3. Bacterial Growth Curve Assay and Time-Kill Assay 

Two separate 96-well plates were used to perform the bacterial growth curve and time-kill assays. Each well was filled with 100 μL of the minimum inhibitory concentration (MIC) of melittin, oxacillin, and cephalothin, as well as a combination containing 25% of the MIC of melittin combined with 25% MIC of oxacillin or cephalothin, all of which diluted in MH broth, the MIC used varied according to the MRSA isolate tested according to the previous MIC results. Subsequently, 100 μL of MRSA isolate inoculum (10^5^ CFU/mL) was added to each well. One of the plates was incubated in the BioTek Epoch 2 (Agilent™, Santa Clara, CA, USA) at 37 °C for 24 h, and the optical density at 600 nm (OD_600_) was measured [14]. The other plate was also incubated at 37 °C for 24 h, and subcultures were prepared at 2, 4, 8, 12, and 24-h time points on MH agar plates. Synergism was determined by considering bacteriostatic effects as a reduction of ≥2 and ≤3 log10 CFU/mL compared to the control, while bactericidal effects were observed when there was a reduction of ≥3log10 CFU/mL compared to the control [13]. 

### 2.4. Effects on Biofilm Formation

MRSA isolates were cultured overnight in Tryptone Soy broth supplemented with 1% glucose and standardized using McFarland’s 0.5 scale. Subsequently, 100 μL of this standardized inoculum was added to each well of a 96-well flat-bottomed plate. In these wells, 100 μL of melittin, oxacillin, and a combination of both (mel+oxa) were also added at concentrations corresponding to 25%, 50%, and 75% of their respective MICs. Following a 48-h incubation period at 37 °C, the plate underwent the following procedures: washing with phosphate-buffered saline (PBS), fixing the biofilm with methanol for 20 min, air-drying the fixed biofilm overnight, staining the biofilm with crystal violet for 15 min, rinsing the plate with water, and resuspending the biofilms in glacial acetic acid (33%) for OD_570_ measurements. Biofilm formation was assessed using the formula: (mean value of treatment/mean value of control) × 100 [15].

### 2.5. Leakage of Proteins and Nucleic Acids

MRSA isolates were centrifuged at 5000× *g* for 15 min, washed three times with PBS, and then incubated at 37 °C with agitation for 4 h using both the MIC and twice the MIC concentration (2× MIC) of melittin, oxacillin, and mel+oxa. Afterward, two milliliters of the suspension were centrifuged at 10,000× *g* for 7 min at 4 °C, and the supernatants were transferred to a 48-well microplate, and the concentrations of released proteins were determined by measuring OD_595_, while the release of nucleic acids was assessed by measuring OD_260_ [16].

### 2.6. Efflux of Potassium and Phosphate Ions

MRSA isolates were exposed to melittin, oxacillin, and mel+oxa at concentrations equal to the MIC and 2× MIC in peptone water at time points 0, 2, and 4 h [17]. The concentrations of potassium and phosphate ions were determined using the MQuant^TM^ Potassium test and MQuant^TM^ Phosphate test, in accordance with the manufacturer’s instructions.

### 2.7. Hemolytic Activity

Heparinized human blood was centrifuged, and a 2% suspension of erythrocytes was prepared in PBS. Melittin, oxacillin, and mel+oxa were diluted in PBS to concentrations of 4 times the minimum inhibitory concentration (4× MIC), 2 times the MIC (2× MIC), MIC, 50% MIC, and 25% MIC. These solutions were then added to a 96-well plate along with 100 µL of the erythrocyte suspension. The plate was incubated at 37 °C for 2 h, followed by centrifugation. Subsequently, 100 µl of the resulting supernatant was transferred to another plate for OD_540_ measurement [18]. The MIC values used were 5.3 μg/mL for melittin, 16 μg/mL for oxacillin, and 2 μg/mL of melittin + 4 μg/mL of oxacillin for mel+oxa.

### 2.8. Cytotoxicity in Keratinocytes by Flow Cytometry

Human epidermal keratinocytes (HaCaT) were cultured at 37 °C with 5% CO_2_ in DMEM supplemented with 10% fetal bovine serum. The cells were treated with the following concentrations: 5.3 and 10.6 μg/mL of melittin for 4 h, oxacillin at 16 and 32 μg/mL for 4 h, melittin at 2 μg/mL + oxacillin at 4 μg/mL for 8 h, and melittin at 4 μg/mL + oxacillin at 8 μg/mL for 8 h. After the specified exposure times, the cells were labeled with Annexin V (FITC) and Propidium Iodide (BD™, NJ, USA). Analysis was performed using a FACSCalibur BD^®^ flow cytometer, with data analysis conducted in Cell Quest Pro PAINT-A-GATE (version 5.1, BD™, Franklin Lakes, NJ, USA).

### 2.9. Statistical Analysis

The experiments were conducted in triplicate, and the data were submitted to a One-Way ANOVA test, followed by the Tukey test. Values of *p* ≤ 0.05 were considered to denote statistical significance and the analyses were performed using GraphPad Prism (version 9.2, Dotmatics™, Bishop’s Stortford, UK).

### 2.10. Protein Extraction and Quantification

Four treatments were evaluated, denoted as follows: C (MRSA control), T1 (MRSA with 25% MIC of melittin, equivalent to 1.3 μg/mL), T2 (25% MIC of oxacillin, equivalent to 4 μg/mL), and T3 (25% of the MIC of mel+oxa, comprising 0.5 μg/mL of melittin with 1 μg/mL of oxacillin). MRSA ATCC was inoculated in MH broth II and incubated at 37 °C for 24 h. Subsequently, the culture was standardized using the 0.5 MacFarland scale, and incubated in Erlenmeyer flasks containing 100 mL of MH broth II at 37 °C for 18 h while subject to the designated treatments.

Fifty milliliters of the culture were centrifuged (15 min, 3220× *g*, 4 °C) and the resulting bacterial pellets were washed with PBS and resuspended in 200 uL of lysis buffer (8 M Urea, 75 mM NaCl, 25 mM Tris-HCl at pH 8, 2 mM MgCl_2_, 1× Roche Protease Inhibitor, and Benzonase 1 U). The samples were submitted to ultrasound Sonics Vibra-Cell VCX-600 Ultrasonic Processor (Artisan Technology Group™, Champaign, IL, USA) were centrifuged (14,000× *g*, 30 min), and the supernatants were collected.

### 2.11. Enzymatic Digestion of Proteins

Fifty micrograms of proteins were diluted in 60 μL of 50 mM ammonium bicarbonate and 25 μL of RapiGest SF Standard surfactant (Waters Corporation™, Milford, MA, USA) was added and incubated at 37 °C/60 min. Samples were submitted to reduction and alkylation, using 10 mM DTT and 45 mM iodoacetamide. For enzymatic digestion, trypsin was used at 1:50 (enzyme: sample) concentration solubilized in ammonium bicarbonate buffer (50 mM, pH 7.8). Hydrolysis took place for 18 h and was stopped using 10 µL of 5% (*v*/*v*) formic acid. Samples were incubated for 90 min at room temperature, centrifuged (14,000× *g*, 6 °C, 30 min), and submitted to Peptide Cleanup C18 columns (Agilent Technologies™, Santa Clara, CA, USA) [19].

### 2.12. Mass Spectrometry

The ultimate 3000 LC liquid nanochromatography system (Dionex™, Germering, Germany), coupled with a Q-Exactive mass spectrometry (Thermo Fisher Scientific™, Bremen, Germany) was employed, the ionization was achieved using a Nanospray ion source (PICOCHIP). The operation mode was positive ionization using Data-Dependent Acquisition (DDA) and mass spectra were acquired from 200 to 2000 (*m*/*z*) at a 70,000 resolution with 100 ms injection time. The fragmentation chamber was conditioned with collision energy between 29 and 35%, 17,500 resolution, 50 ms of injection time, 4.0 *m*/*z* of isolation window, and a dynamic exclusion of 10s. Spectrometry data were acquired using Thermo Xcalibur software (version 4.0.27.19, Thermo Fisher Scientific) [19].

### 2.13. Analysis of Mass Spectrometry Data

PatternLab software (version 4.0.0.84, Fiocruz, Curitiba, Brazil) was employed to identify proteins using the UNIPROT database (Taxonomy *Staphylococcus aureus* MRSA—22,492 sequences on 26 December 2021); using the following parameters: trypsin enzyme; allowance for 2 missed cleavages; post-translational modification of carbamidomethylation for cysteine residues, and variable oxidation for methionine residues; MS and MS/MS tolerance errors of 0.0200 ppm; with a maximum false discovery rate (FDR) ≤ 1%. MetaboAnalyst 4.0^®^ software was utilized for T-student and Fold-change analyses (*p* < 0.05; Fc ≥ 1.5). Additionally, the STRING (https://string-db.org/ (accessed on 13 August 2023)) was used to perform the protein–protein interactions network. 

## 3. Results

### 3.1. Minimum Inhibitory Concentration (MIC) and Minimum Bactericidal Concentration (MBC)

Melittin exhibited antibacterial activity, with a concentration of 5.3 μg/mL against ATCC MRSA and 4.0 μg/mL against the isolate. Moreover, it demonstrated bactericidal effects at a concentration of 8.0 μg/mL against both isolates. The ATCC strain exhibited a minimum inhibitory concentration (MIC) of 16.0 μg/mL for both oxacillin and cephalothin, while the isolate showed MIC values of 8.0 μg/mL for oxacillin and 2.0 μg/mL for cephalothin. Both isolates exhibited the same minimum bactericidal concentration (MBC) of 16 μg/mL for oxacillin. The MBC for the ATCC strain was 32 μg/mL, while for the isolate, it was 2.0 μg/mL (Table 1).

### 3.2. Bacterial Growth Curve Assay and Time-Kill Assay

Melittin exhibited rapid inhibitory action, achieving bactericidal effects within 2 h for ATCC MRSA and 4 h for the isolate MRSA. In contrast, antibiotics displayed slower inhibitory activities, taking 6 h for oxacillin and 8 h for cephalothin to achieve bactericidal effects against both isolates. The combination of melittin and cephalothin demonstrated a synergistic bactericidal effect after 8 h, exclusively against the isolate MRSA, while it exhibited an antagonistic effect for ATCC MRSA. Moreover, the mel+oxa combination displayed synergistic bactericidal activity after 6 h for both isolates (Figure 1).

### 3.3. Effects on Biofilm Formation

The association of oxacillin with melittin demonstrated inhibition of MRSA biofilms. The 75% of the minimum inhibitory concentration (MIC) of oxacillin and mel+oxa exhibited the highest inhibition, approximately 65%, in biofilm formation for both isolates. Melittin alone showed biofilm inhibition exclusively on the MRSA isolate (Figure 2).

### 3.4. Leakage of Proteins and Nucleic Acids

After a four-hour treatment with melittin, oxacillin, and mel+oxa, both at MIC and 2× MIC concentrations, proteins (Figure 3) and nucleic acids leakage occurred (Figure 4). For the ATCC strain, the mel+oxa at MIC exhibited the most significant protein and nucleic acid leakage, releasing 0.040 μg/μL (±0.001) of proteins and displaying an OD_260_ of 0.131 (±0.008) for nucleic acids. The mel+oxa at 2× MIC resulted in a leakage of 0.046 μg/μL (±0.001) of proteins and an OD_260_ of 0.141 (±0.010) for nucleic acids. Meanwhile, for the MRSA isolate, at 2× MIC concentrations, all treatments led to greater protein leakage, ranging from 0.034 to 0.043 μg/μL, but mel+oxa (both MIC and 2× MIC) induced the most substantial nucleic acid leakage, with OD_260_ values ranging from 0.130 to 0.142.

### 3.5. Efflux of Potassium and Phosphate Ions

In both MRSA isolates, following a 2-h exposure to 2× MIC concentrations of all treatments, there was a leakage of 250 mg/L of potassium and 10 mg/L of phosphate. Specifically, the MIC of melittin caused a phosphate leakage of 10 mg/L in both isolates, along with a potassium leakage of 250 mg/L in the case of isolate MRSA. Subsequently, after 4 h of exposure, both MIC and 2× MIC concentrations of all treatments resulted in the release of 250 mg/L of potassium and 10 mg/L of phosphate for both isolates (Table 2).

### 3.6. Hemolytic Activity

At a concentration of 4× MIC, melittin (21 μg/mL) and mel+oxa (8 μg/mL melittin + 16 μg/mL oxacillin), induced 100% hemolysis of erythrocytes. Melittin caused 4.2% hemolysis at a 2× MIC concentration (10.6 μg/mL), and this percentage decreased to 0.2% at MIC and 25% MIC, indicating no hemolytic activity. Notably, mel+oxa did not induce hemolysis at concentrations ranging from 2× MIC to 25% MIC, and oxacillin showed no hemolytic activity across all tested concentrations (Table 3).

### 3.7. Cytotoxicity in Keratinocytes by Flow Cytometry

Flow cytometry analysis revealed that melittin, oxacillin, and mel+oxa, both at MIC and 2× MIC concentrations, did not induce apoptosis or late cell death in HaCaT cells (Figure 5). When HaCaT cells were incubated with all treatments for 4 and 8 h, the percentage of apoptosis observed ranged from 1.06% to 6.11% (Table 4).

### 3.8. Proteomic Analysis

Proteomic analysis allowed the identification of 72 proteins in the control group, 96 in the melittin treatment (with 68 proteins in common with the control), 43 in the oxacillin treatment (with 28 proteins in common with the control), and 112 proteins in the melittin-oxacillin treatment (with 67 proteins in common with the control) (Figure 6). According to the Volcano Plot analysis, 17 proteins exhibited differential expression, either upregulated or downregulated, across all treatments (melittin, oxacillin, and mel+oxa) in comparison to the control group. Furthermore, the Vip Scores analysis highlighted 33 more relevant proteins (Table 5).

All treatments altered the MRSA protein profile, and the expressed proteins were categorized based on their molecular function and biological process (Figure 7). The molecular functions that exhibited the most significant changes were binding, particularly binding to ions, ATP, and/or nucleic acids. This was followed by proteins with catalytic activity, mainly involving transferase and oxidoreductase activities, as well as structural components such as ribosomes and cell membranes. In terms of biological processes, the most pronounced alterations were observed in bacterial metabolism, which affected translation, protein synthesis, and energy metabolism.

Melittin treatment led to a decrease in the expression of proteins associated with protein synthesis, including ribosome proteins 30S S3, 50S L9, 50S L11, which are involved in the translation process, as well as the elongation factor G. Additionally, this treatment resulted in the downregulation of pyruvate kinase, serine hydroxymethyltransferase, and chaperone protein DnaK, which are related to glycolysis, one-carbon metabolism, and stress response, respectively. Furthermore, proteins linked to energy metabolism exhibited increased expression, such as 6-phosphogluconate dehydrogenase, decarboxylating (6PGDH), aldehyde-alcohol dehydrogenase, and enolase. Aspartate–tRNA ligase and thymidylate synthase also showed increased expression and were associated with protein and DNA biosynthesis, respectively.

Oxacillin treatment resulted in the downregulation of proteins associated with energy metabolism, including 6-phospho-beta-galactosidase and ATP synthase subunit beta, while increasing the expression of aldehyde-alcohol dehydrogenase, enolase, and glyceraldehyde-3-phosphate dehydrogenase. Additionally, this treatment led to the upregulation of ribosomal proteins 30S S19 and 50S L10, with lower expression observed for 30S S2 and 50S L9. Proteins related to carbon metabolism, S-adenosylmethionine synthase and serine-hydroxymethyltransferase, were downregulated, as were cell membrane proteins, including extracellular matrix-binding protein ebh and ATP synthase subunit beta. Notably, cold shock protein CspA, involved in the response to stress, showed an upregulation in response to the treatment.

In the mel+oxa treatment, all biological processes (metabolism, cellular functions, reproduction, cellular localization, and response to stimuli) were affected, with the most pronounced impact observed in protein synthesis, followed by energy metabolism, which exhibited the highest protein expression changes. This treatment brought about alterations in proteins associated with various functions, including energy metabolism proteins, such as 6PGDH, enolase, and formate acetyltransferase; transport-related proteins, including ABC transporters, ATP-binding proteins, phosphoenolpyruvate-protein phosphotransferase, PstB; proteins involved in protein synthesis, like ribosomal proteins 50S L11 and 50S L18, DAHP synthetase-chorismate mutase, ribosomal RNA small subunit methyltransferase H, and ribosome-binding ATPase YchF; proteins related to purine metabolism, such as GMP reductase and inosine-5′-monophosphate dehydrogenase; S-adenosylmethionine synthase, associated with carbon metabolism; and a cell division protein, FtsZ (Figure 8).

## 4. Discussion

Antibiotic synergy with another antimicrobial can affect different bacterial targets, aiding in the overcoming of resistance mechanisms, enhancing bactericidal activity, reducing antimicrobial doses, minimizing toxicity, and restoring effectiveness [20]. The combination of melittin and oxacillin has demonstrated synergistic and bactericidal effects against MRSA, allowing for the use of lower antimicrobial doses while maintaining bactericidal activity.

Melittin exhibits a fast inhibitory effect against vancomycin-resistant *S. aureus* (VRSA) isolates, demonstrating bactericidal activity within a timeframe ranging from 15 min to 3 h [21]. Additionally, synergistic interactions between melittin and doripenem, a β-lactam, have proven effective against *Acinetobacter baumannii* and *Pseudomonas aeruginosa* [22]. Furthermore, a previous study has already established the bactericidal synergism of melittin combined with oxacillin (mel+oxa) against MRSA [13].

The synergistic action of AMPs and antibiotics offers a promising strategy for overcoming biofilm resistance, enhancing antibiofilm activity, and potentially delaying the emergence of resistance [23]. In combating MRSA biofilm formation, melittin has demonstrated synergistic effects when combined with vancomycin, rifampicin [24], as well melittin synergism with oxacillin. However, it is noteworthy that melittin alone did not exhibit inherent antibiofilm activity, possibly due to the use of subinhibitory concentrations and the strong biofilm production capabilities of the tested MRSA isolates. Biofilms, characterized by bacteria within an extracellular polymeric matrix, display heightened resistance compared to planktonic cells, owing to limited antibiotic diffusion, constrained growth and metabolism, the presence of persistent cells, upregulated resistance mechanisms, and the potential for mutations and horizontal gene transfer [23].

Melittin, oxacillin, and their association resulted in an increased leakage of proteins, nucleic acids, and the efflux of potassium and phosphate ions, indicating an impact on cell membrane permeability. The metabolites produced by *Pediococcus pentosaceus* 4I1 demonstrated inhibitory effects on both *S. aureus* and *E. coli*, with evidence of membrane disruption substantiated by elevated potassium efflux, nucleic acid leakage, and the increase of electrical conductivity [17].

Melittin interacts with negatively charged bacterial membranes, leading to depolarization, permeabilization, and eventual rupture. The specific mechanism of interaction with membranes can vary depending on the composition of the lipid bilayer and the quantity of melittin bound or inserted into the surface. Melittin has the capacity to influence membranes either by forming pores or by adopting a carpet-like mode of action [25]. Melittin was observed to induce pore formation in the cytoplasmic membrane of *Acinetobacter baumannii*, likely attributable to its hydrophobic N-terminal and central regions. Once melittin establishes binding, others can more readily adhere through hydrophobic bonding, leading to their accumulation on the membrane and ultimately facilitating rupture or pore formation [14].

Melittin exhibited bactericidal efficacy in mice with third-degree burns infected by VRSA at a dose of 32 μg/mL within the lesions. Importantly, melittin application at burn sites and on the skin of healthy mice did not manifest cytotoxic or hemolytic effects, underscoring its potential to eliminate VRSA without inducing harmful cytotoxicity [14,21]. Furthermore, the combination of melittin with doripenem resulted in reduced cytotoxicity in normal kidney cells (HEK293) [22]. Similarly, when melittin was synergistically employed with rifampicin and vancomycin against methicillin-resistant *Staphylococcus epidermidis* (MRSE), it did not trigger hemolysis in human erythrocytes or provoke cytotoxicity in HEK293 cells [26]. The challenge associated with the use of AMPs often lies in their potential for hemolytic activity and cytotoxicity [23]. However, melittin in combination with oxacillin (mel+oxa), in the tested concentrations, did not exhibit hemolytic activity in human erythrocytes and apoptosis in HaCaT cells, thereby demonstrating enhanced effectiveness, reduced toxicity, and the potential to mitigate the development of resistance mechanisms.

Antimicrobial peptides (AMPs) have the capability to influence the integrity of bacterial cell membranes as well as other intracellular targets, potentially affecting processes such as DNA synthesis, protein synthesis, cell wall synthesis, and various metabolic pathways [6]. Notably, when MRSA is treated with melittin, it induces alterations in the protein synthesis process, impacts energy metabolism, disrupts one-carbon metabolism, interferes with DNA synthesis, and triggers stress responses.

Elongation factor G (EF-G) plays a pivotal role in the translocation of messenger RNAs (mRNA) and transfer RNAs (tRNA) within ribosomes, and it is indispensable for protein synthesis, rendering it a potential target for antibacterial agents [27]. Bacterial translation necessitates the assembly of a 70S macromolecular complex, comprising the 30S small ribosomal subunit, typically consisting of 21 ribosomal proteins (S1–S21), and the 50S large subunit, typically composed of 33 ribosomal proteins (L1–L36) [28]. When MRSA is subjected to melittin treatment, a notable reduction in the expression levels of most ribosomal proteins and EF-G is observed, indicative of its impact on protein synthesis. Conversely, the expression of Aspartate–tRNA ligase (AspRS) is heightened, underscoring its importance in the protein synthesis process [29].

The enzyme 6PGDH plays a crucial role in the pentose-phosphate pathway, which is closely linked to carbohydrate metabolism and serves as an alternative to glycolysis. In the context of melittin treatment, it exhibited upregulation, potentially facilitating the elimination of reactive oxygen species and the synthesis of nucleotides for repairing DNA damage [30]. Aldehyde-alcohol dehydrogenase (ADHE), involved in alcohol synthesis and a participant in energy metabolism, also demonstrated upregulation [31]. Enolase, which plays a key role in the glycolysis process and has associations with pathogenicity in *S. aureus* while being a part of the messenger RNA degradosome complex [32], exhibited upregulation across all treatments.

The increased expression of proteins associated with energy metabolism may signify a heightened demand for energy by bacteria to ensure their survival [33]. In the case of melittin treatment, there was an increase in the expression of certain energy metabolism proteins (6PGDH, ADHE, and enolase), except for pyruvate kinase (PK), suggesting a potential survival strategy employed by MRSA. Pyruvate kinase plays a crucial role in carbohydrate metabolism, catalyzing the final step in glycolysis, where phosphoenolpyruvate is converted to pyruvate, accompanied by ADP phosphorylation to ATP. Additionally, PK is involved in the tricarboxylic acid cycle, which has implications for bacterial pathogenicity [34]. Notably, melittin downregulated PK, potentially interfering with glycolysis and pyruvate production.

In the presence of melittin treatment, the expression of serine hydroxymethyltransferase (SHMT) was reduced, while thymidylate synthase (TS), crucial for DNA synthesis and bacterial replication, exhibited an increased expression, potentially as part of a survival mechanism. Both SHMT and TS are integral components of the one-carbon metabolism pathway, essential for the biosynthesis of purines and thymidines, and play a pivotal role in promoting cell growth and proliferation [35].

Furthermore, chaperones play a significant role in the stress response by assisting in the folding of proteins, preventing protein aggregation, and aiding in the repair of proteins that have been damaged by stress [36]. In the presence of melittin, the chaperone protein DnaK experienced downregulation, consequently impacting the bacterial cell cycle and suppressing the expression of stress response proteins. This downregulation of DnaK allows antimicrobials to exert their action more effectively.

Treatment with oxacillin resulted in reduced expression of cell membrane proteins, including the extracellular matrix-binding protein ebh and the ATP synthase subunit beta. These proteins are associated with energy metabolism, with the ATP synthase subunit beta specifically playing a crucial role in ATP synthesis and transmembrane transport, thereby being vital for the growth and metabolism of *S. aureus* [33]. The presence of cold shock proteins indicates a response to the stress induced by oxacillin. Overall, oxacillin treatment induced alterations in proteins involved in energy metabolism, protein synthesis, cellular localization, and stress response.

Cell division protein FtsZ polymerizes within bacterial cells, forming a ring known as the Z ring, which precisely marks the site for cell division. This ring recruits other essential proteins, contributing to the formation of the divisome complex. Alterations in FtsZ can impede cell division, ultimately leading to cell death, making it a promising target for novel antibacterial agents [37]. In the context of mel+oxa treatment, FtsZ displayed reduced expression, highlighting the potential efficacy of this synergistic approach against MRSA. The agent TXA707, when combined with β-lactams, has demonstrated an impact on FtsZ, creating an opportunity for β-lactams to target PBP2 in MRSA. Since the functions of PBPs are regulated by the cell cycle, FtsZ plays a crucial role in locating PBPs during the cell division process [38]. Mel+oxa treatment influenced FtsZ, possibly causing rapid responses in bacterial cells with respect to nutrient availability, thereby channeling a greater portion of energy towards protein synthesis [29]. Additionally, formate acetyltransferase, linked to energy support, exhibited upregulation and is capable of operating when pyruvate is available, potentially becoming more pronounced in anaerobic conditions [39].

S-adenosylmethionine synthase helps in one-carbon metabolism, which is important for cell growth and development. It actively participates in the synthesis of purines and pyrimidines, as well as the metabolism of amino acids and nucleotides [40]. Additionally, the ribosome-binding ATPase YchF is linked to protein degradation and the regulation of translation during stress responses [41]. Furthermore, ribosomal RNA small subunit methyltransferase H, which is responsible for the processing of ribosomal RNAs and promotes RNA methylation using S-adenosylmethionine [42], was also impacted by mel+oxa treatment, underscoring its influence on translation and protein synthesis.

DAHP synthetase and chorismate mutase (CM) are integral components of the shikimate pathway, a series of enzymatic reactions crucial for protein synthesis. Notably, this pathway is unique to bacteria and is absent in humans, rendering it an attractive target for antibacterial agents. Typically, DAHP is subject to feedback inhibition in the presence of aromatic amino acids [43]. In the mel+oxa treatment, there was an observed overexpression of DAHP synthetase and CM, hinting at a possible deficiency of aromatic amino acids within MRSA. Additionally, GMP reductase and inosine-5′-monophosphate dehydrogenase participate in purine synthesis, a process vital for DNA and RNA synthesis, ultimately promoting bacterial growth [44]. 

ABC transporters facilitate ATP-dependent transport, with the capability to transport a diverse range of substrates, including sugars, ions, peptides, antimicrobial agents, translation factors, and others. They promote energy production, aiding DNA repair processes, and contributing to virulence [45,46]. In response to mel+oxa treatment, these transporters exhibit upregulation, potentially as an attempt to expel the treatment. Phosphoenolpyruvate-protein phosphotransferase primarily functions in sugar transport systems, while PstB plays a significant role in phosphate transport and ATP hydrolysis [45,46].

Nisin, an antimicrobial peptide synthesized by Lactococcus lactis, demonstrated synergistic effects against MRSA when combined with oxacillin. Its mechanisms of action were found to be closely linked to protein synthesis and energy metabolism [47], akin to the actions observed with mel+oxa against MRSA. The combination of melittin and oxacillin not only influenced MRSA’s cellular structure, including the membrane and cell wall, but also had the potential to interfere with translation, thereby affecting various vital biological processes such as protein synthesis, energy metabolism, cell division, transport processes, metabolism of purines, and DNA synthesis. This underscores the broad impact of this synergism on essential MRSA proteins involved in diverse biological processes.

## 5. Conclusions

Melittin exhibited bactericidal against MRSA, disrupting the integrity of the bacteria cell membrane, and inducing changes in its energy metabolism and protein synthesis. Melittin association with oxacillin displayed a synergistic and bactericidal effect in MRSA. Furthermore, this combination inhibited MRSA biofilm formation, without demonstrating hemolytic activity on human erythrocytes, and apoptosis or necrosis on human keratinocytes (HaCaT) at the MIC (2 μg/mL of melittin + 4 μg/mL of oxacillin) and 2× MIC (4 μg/mL of melittin + 8 μg/mL of oxacillin) concentrations. This association also had impact on MRSA cell membrane permeability and affected essential proteins related to energy metabolism, protein synthesis, cell division, transport, and DNA synthesis. The advantage of melittin association with oxacillin lies in their distinct mechanisms of action, potentially reducing dosages, minimizing adverse effects, and mitigating the development of bacterial resistance. This approach targets various biological processes within MRSA and holds promise as a candidate for the development of antibacterial drugs against MRSA.

## Figures and Tables

**Figure 1 microorganisms-11-02868-f001:**
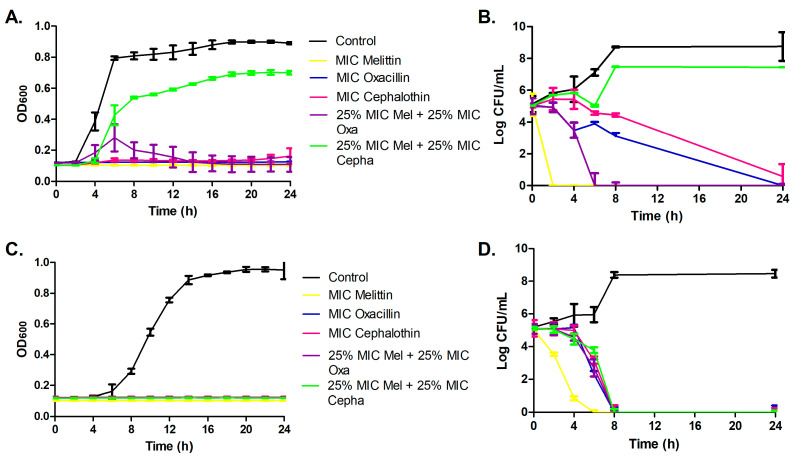
Curve of the inhibitory activity (OD_600_) for ATCC MRSA (**A**) and isolate MRSA (**C**) and time-kill curve of the bactericidal activity, in Log CFU/mL, for ATCC MRSA (**B**) and isolate MRSA (**D**).

**Figure 2 microorganisms-11-02868-f002:**
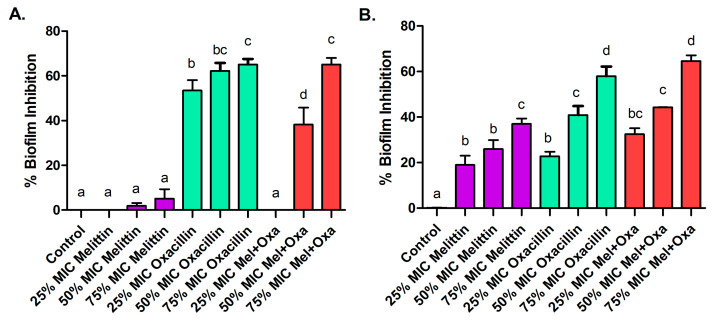
The percentage (%) of biofilm inhibition at subinhibitory concentrations of 25%, 50%, and 75% MIC for melittin, oxacillin, and the melittin-oxacillin combination against ATCC MRSA (**A**) and MRSA isolate (**B**). Distinct letters represent significant differences in percentage of biofilm inhibition among the tested treatments when *p* ≤ 0.05.

**Figure 3 microorganisms-11-02868-f003:**
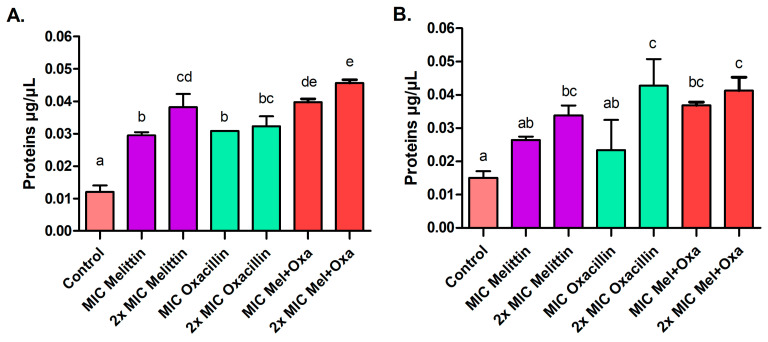
The protein leakage, measured in μg/mL, after a 4-h exposure of both ATCC MRSA (**A**) and MRSA isolate (**B**) to treatments with MIC and 2× MIC of melittin, oxacillin, and the melittin and oxacillin combination. Distinct letters represent significant differences in leakage of proteins among the tested treatments when *p* ≤ 0.05.

**Figure 4 microorganisms-11-02868-f004:**
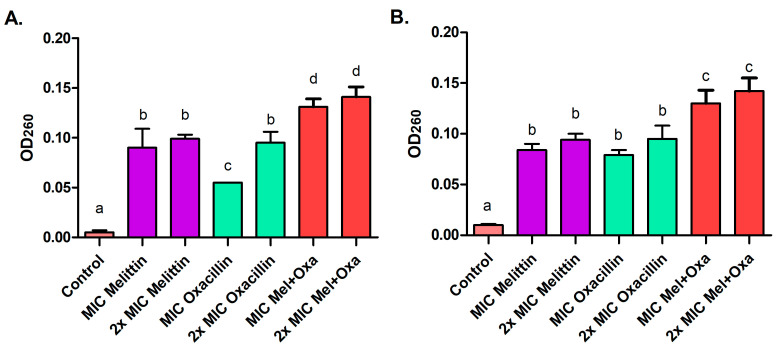
The release of nucleic acids, measured at an optical density of 260 nm, after a 4-h exposure of both ATCC MRSA (**A**) and MRSA isolate (**B**) to treatments with MIC and 2× MIC of melittin, oxacillin, and the melittin and oxacillin combination. Distinct letters represent significant differences in release of nucleic acids among the tested treatments when *p* ≤ 0.05.

**Figure 5 microorganisms-11-02868-f005:**
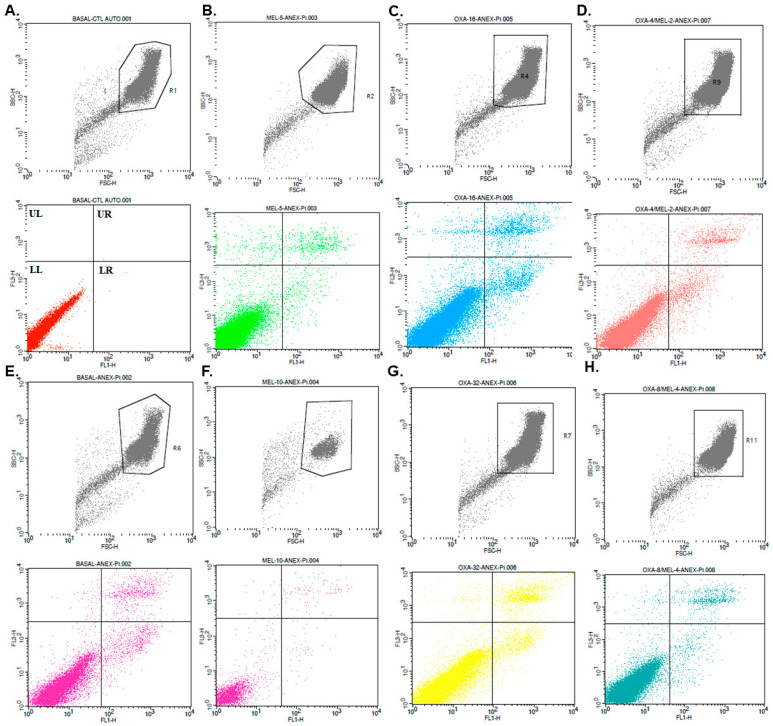
Analysis of HaCaT cells was conducted following treatment with melittin, oxacillin, and their combination (mel+oxa) to assess the induction of early apoptosis and late apoptosis/necrosis. The evaluation employed annexin V labeling (FITC) and propidium iodide. The treatments were represented as follows: (**A**) Basal negative control (autofluorescence); (**B**) MIC Melittin; (**C**) MIC Oxacillin; (**D**) MIC Mel+Oxa; (**E**) Basal negative control; (**F**) 2× MIC Melittin; (**G**) 2× MIC Oxacillin; (**H**) 2× MIC Mel+Oxa (Quadrant UL: represents late apoptosis or necrotic cells—anexxin V negative and propidium iode positive; Quadrant UR: represents late apoptotic or necrotic cells—anexxin V positive and propidium iodide positive; Quadrant LL: represents live cells—anexxin V negative and propidium iodide negative; Quadrant LR: represents early apoptotic cells—anexxin V positive and propidium iodide negative).

**Figure 6 microorganisms-11-02868-f006:**
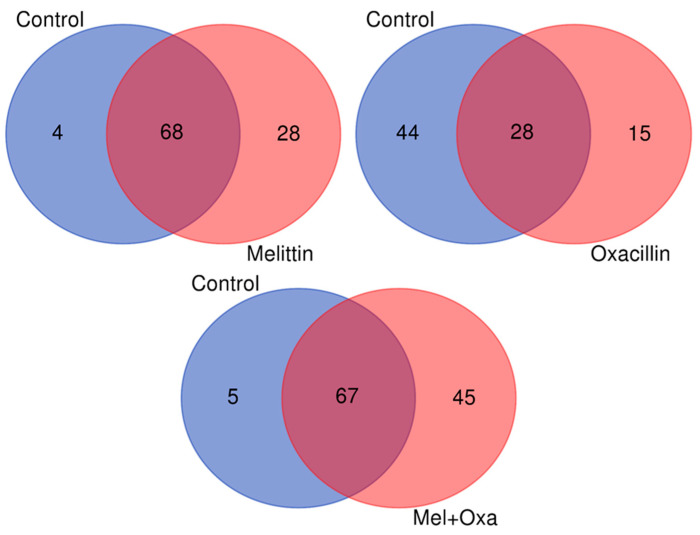
Venn diagram of all proteins identified in the proteomic analysis of the three treatments (melittin, oxacillin, and mel+oxa) in comparison to the proteins identified in the control group.

**Figure 7 microorganisms-11-02868-f007:**
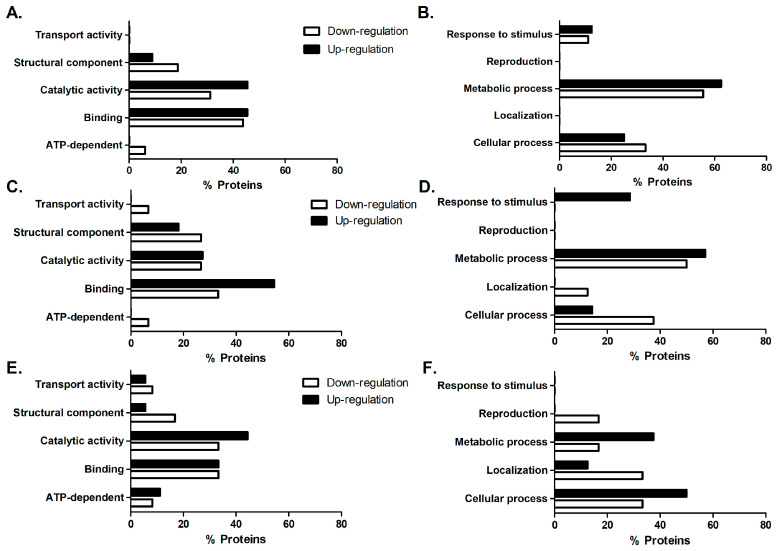
The molecular function of subinhibitory treatments with melittin (**A**), oxacillin (**C**), and mel+oxa (**E**), as well as the biological processes influenced by the treatments with melittin (**B**), oxacillin (**D**), and mel+oxa (**F**) in the differentially expressed proteins of MRSA when exposed to the treatments, considering both upregulation and downregulation, and compared to the control group.

**Figure 8 microorganisms-11-02868-f008:**
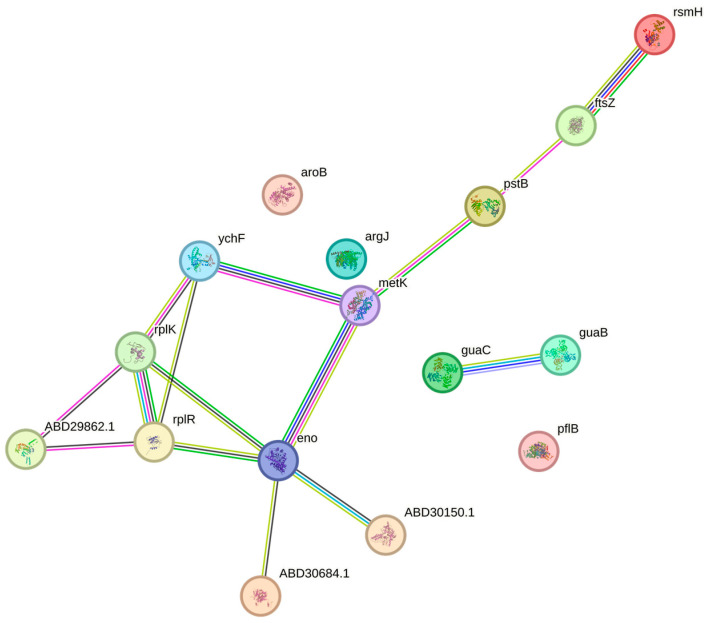
Protein–protein interactions networks for the most relevant proteins influenced by the treatment melittin in association with oxacillin (mel+oxa) against MRSA. Strong associations between proteins are depicted by connecting lines. rsmH: S-adenosyl-methyltransferase, ftsZ: Cell division protein FtsZ, pstB: Phosphate ABC transporter ATP-binding protein, metK: S-adenosylmethionine synthetase, eno: Enolase, ABD30684.1: 6-phosphogluconate dehydrogenase, decarboxylating, ABD30150.1: Phosphoenolpyruvate-protein phosphotransferase, ABD29862.1: ABC transporter, ATP-binding protein, ychF: Ribosome-binding ATPase YchF, rplR: Ribosomal protein L18, rplK: Ribosomal protein L11, aroB: DAHP synthetase, guaB: inosine-5′-monophosphate dehydrogenase, guaC: Guanosine monophosphate reductase, argJ: N-acetyltransferase, pflB: Formate acetyltransferase.

**Table 1 microorganisms-11-02868-t001:** Minimum inhibitory concentration (MIC) and minimum bactericidal concentration (MBC), expressed in µg/mL, of melittin, oxacillin, and cephalothin against MRSA.

MRSA	Melittin		Oxacillin		Cephalothin	
	MIC	MBC	MIC	MBC	MIC	MBC
ATCC	5.3	8.0	16.0	16.0	16.0	32.0
Isolate	4.0	8.0	8.0	16.0	2.0	2.0

**Table 2 microorganisms-11-02868-t002:** The efflux of potassium and phosphate ions, measured in mg/L, after the treatment of both ATCC MRSA and the isolate with melittin, oxacillin, and mel+oxa, at MIC and 2× MIC concentrations, with the exposure times of 0 h (T0h), 2 h (T2h), and 4 h (T4h).

	Potassium Ions						Phosphate Ions					
	**T0h**		**T2h**		**T4h**		**T0h**		**T2h**		**T4h**	
Control	0		0		0		0		0		0	
MIC	1×	2×	1×	2×	1×	2×	1×	2×	1×	2×	1×	2×
Melittin	0	0	0/250 *	250	250	250	0	0	10	10	10	10
Oxacillin	0	0	0	250	250	250	0	0	0	10	10	10
Mel+Oxa	0	0	0	250	250	250	0	0	0	10	10	10

* Efflux of 0 mg/L and 250 mg/L of potassium ions for MRSA ATCC and MRSA isolate, respectively.

**Table 3 microorganisms-11-02868-t003:** The percentage (%) of hemolysis in human red cells after the treatments with melittin, oxacillin, and mel+oxa at different concentrations (4× MIC, 2× MIC, MIC, 50% MIC, and 25% MIC).

	4× MIC	2× MIC	MIC	50% MIC	25% MIC
Melittin	100 ± 0.9 ^aA^	4.2 ± 7.2 ^bA^	0.2 ± 0.2 ^cA^	0.2 ± 0.4 ^cA^	0.2 ± 0.5 ^cA^
Oxacillin	0 ± 0.8 ^aB^	0 ± 0.5 ^aB^	0 ± 0.2 ^aA^	0 ± 0.0 ^aA^	0 ± 0.0 ^aA^
Mel+Oxa	100 ± 3.8 ^aA^	0 ± 0.2 ^bB^	0 ± 0.5 ^bA^	0.0 ± 0.1 ^aA^	0.0 ± 0.0 ^aA^

Distinct lowercase letters in the same row represent significant differences in hemolytic activity among the tested concentrations of the same product when *p* ≤ 0.05, while distinct capital letters in the columns denote significant differences in hemolytic activity between the tested products when *p* ≤ 0.05.

**Table 4 microorganisms-11-02868-t004:** The percentage of the viable cell population, late apoptotic/necrotic, and early apoptotic HaCaT cells after exposure to MIC and 2× MIC concentrations of melittin, oxacillin, and mel+oxa.

Treatments	UL (Late Apoptotis/Necrosis) %	UR (Late Apoptosis/ Necrosis) %	LL (Live Cells) %	LR (Early Apoptosis) %
Basal negative control (autofluorescence)	0.00	0.01	99.97	0.02
Basal negative control	0.50	5.40	87.99	6.11
MIC Melittin	0.77	1.64	96.52	1.06
2× MIC Melittin	0.78	3.94	93.62	1.66
MIC Oxacillin	0.89	3.95	90.88	4.29
2× MIC Oxacillin	0.55	5.94	89.42	4.10
MIC Mel+Oxa	0.21	3.53	92.22	4.04
2× MIC Mel+Oxa	0.36	2.54	95.65	1.44

**Table 5 microorganisms-11-02868-t005:** Upregulation and downregulation of MRSA proteins in response to the treatments T1—Melittin, T2—Oxacillin, T3—Melittin with Oxacillin, in comparison to the C—Control (MRSA without treatment) according to the Volcano Plot and Vips Scores analysis.

Proteins	Molecular Function	Biological Process	Cellular Component	Protein Expression			
				C	T1	T2	T3
30S ribosomal protein S2 *	Structural component; Binding—nucleic acid	Translation	Ribosome	✓		down	
30S ribosomal protein S3 *	Structural component; Binding—nucleic acid	Translation	Ribosome	✓	down		
30S ribosomal protein S19 *	Structural component; Binding—nucleic acid	Translation	Ribosome	✓	up	up	
50S ribosomal protein L9 *	Structural component; Binding—nucleic acid	Translation	Ribosome		down	down	
50S ribosomal protein L10	Structural component; Binding—nucleic acid	Translation	Ribosome			up	
50S ribosomal protein L11 *	Structural component; Binding—nucleic acid	Translation	Ribosome	✓	down		down
50S ribosomal protein L18	Structural component; Binding—nucleic acid	Translation	Cytoplasm; Ribosome				up
6-phospho-beta-galactosidase *	Glycosidase; Hydrolase	Lactose catabolic process via	–	✓		down	
6-phosphogluconate dehydrogenase, decarboxylating *	Binding; Oxidoreductase	Gluconate utilization; Pentose phosphate pathway	–	✓	up		up
ABC transporter, ATP-binding protein	Binding—ATP; Hydrolase; Transport	Carbohydrate transport	–				up
Aldehyde-alcohol dehydrogenase *	Binding; Oxidoreductase	Metabolic process of alcohol; Carbon utilization	–	✓	up	up	
Aspartate–tRNA ligase	Binding—nucleic acid and ATP; Ligase activity	Protein biosynthesis	Cytoplasm	✓	up		
ATP synthase subunit beta	ATP-dependent; Ligase activity; Transport	ATP synthesis; Transport	Cell membrane	✓		down	
Cell division protein FtsZ	Binding; Hydrolase	Cell division	Cytoplasm; Cell division site				down
Chaperone protein DnaK	ATP-dependent; Binding; Chaperone; Hydrolase	Stress response	–	✓	down		
Cold shock protein CspA	Structural component; Binding—nucleic acid	Stress response	Cytoplasm			up	
DAHP synthetase-chorismate mutase	Lyase activity	Amino acid biosynthesis aromatics; Chorismate metabolic process	–				up
Elongation factor G	Translation factor; Hydrolase	Protein biosynthesis	Cytoplasm	✓	down		
Enolase *	Binding; Lyase activity	Glycolysis	Extracellular region; Cell surface		up	up	up
Extracellular matrix-binding protein ebh	Structural component	–	Cell membrane	✓		down	
Formate acetyltransferase	Transferase activity	Glucose metabolic process	Cytoplasm				up
Glyceraldehyde-3-phosphate dehydrogenase	Binding; Oxidoreductase	Glucose metabolic process	–	✓		up	
GMP reductase	Oxidoreductase	Nucleotide metabolic process	Macromolecular complex				up
Inosine-5’-monophosphate dehydrogenase *^#^	Binding; Oxidoreductase	GMP biosynthesis	–	✓			down
N-acetyltransferase *	Transferase activity	–	–	✓	down	down	down
Phosphoenolpyruvate-protein phosphotransferase	Transferase activity	Transport	Cytoplasm	✓			down
PstB	ATP-dependent; Binding—ATP	Phosphate transport	Cell membrane				down
Piruvate kinase	Binding—ATP; Transferase activity	Glycolysis	–		down		
Ribosomal RNA small subunit methyltransferase H	Binding; Transferase activity	rRNA processing	–				down
Ribosome-binding ATPase YchF	ATP-dependent; Binding—ATP and ions; Hydrolase	–	–				up
S-adenosylmethionine synthase	Binding—ATP; Transferase activity	One-carbon metabolism	Cytoplasm	✓		down	up
Serine hydroxymethyltransferase *	Binding; Transferase activity	One-carbon metabolism; Amino acid biosynthesis	Cytoplasm	✓	down	down	
Thymidylate synthase *	Transferase activity	Nucleotide biosynthesis	Cytoplasm		up		
Uncharacterized protein	–	–	–			up	

The symbols convey the following meanings: ✓ signifies presence; “down” indicates downregulation in the treatment, and “up” signifies upregulation in the treatment. The symbol: * indicted proteins that are underlined represent those that exhibit differential expression based on the Volcano Plot analysis and the symbol: ^#^ indicated the only protein exclusively present in the Volcano Plot and do not appear in the Vip Scores.

## Data Availability

Mass spectrometry data have been uploaded to the MassIVE Repository from Computer Science and Engineering University of California, San Diego (https://massive.ucsd.edu/ProteoSAFe/dataset.jsp?task=47d4f410165d4ab19b872bf391928182, accessed on 16 November 2023). AF1 represents the control group with only MRSA, while the remaining groups involve MRSA with the treatments: AF2 (oxacillin), AF3 (melittin), and AF4 (melittin combined with oxacillin).

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
