# Peer review of "Synergistic Antibacterial Efficacy of Melittin in Combination with Oxacillin against Methicillin-Resistant *Staphylococcus aureus* (MRSA)"

_microorganisms, 2023, doi:10.3390/microorganisms11122868_

Round 1
Reviewer 1 Report
Comments and Suggestions for Authors
The manuscript titled “Synergistic antibacterial efficacy of melittin in combination with oxacillin against methicillin-resistant Staphylococcus aureus (MRSA)” written by Pereira et al. focuses on the synergistic effect of the AMP melittin associated with oxacillin against MRSA. The topic of the antibiotic resistance is very current. In the last years many reaserches had the aim to develop new strategies able to overcome this global threat. The manuscript is well organized and the resullts reported could be of interest for the scientific community.
Only minor revisions are required before the pubblication in the journal:
- Introduction: line 44 needs a reference (e.g. International Journal of Molecular Sciences, 2023, 24(5), 4872). Some data on the known antibacterial and anti-biofilm properties of the AMP melittin should be reported (add the reference Nat Prod Res. 2022 Dec;36(24):6381-6388). Authors should report in the introduction Melittin’s antibacterial and antibiofilm activity alone and/or in combination with gentamicin, ciprofloxacin, rifampin, and vancomycin on biofilm-forming MDR-P. aeruginosa and MDR-MRSA strains ( Front. Microbiol. 14:1030401. doi: 10.3389/fmicb.2023.1030401). At line 53 add the recent reference on the antibiotic resistance of biofilm: Current Medicinal Chemistry, 2022, 29(25), pp. 4307–4310
- Line 133: correct CO2
- Table 1: the concentration must be specified (µg/mL)
- Figures 5 and 7 are not very clear, enhance their quality and the readability
- Line 403: correct “antibactericidal” in “bactericidal” or “antimicrobial”
- Ref 22: the year should be in bold
Author Response
Manuscript ID: microorganisms-2673353
Type of manuscript: Article
Title: Synergistic antibacterial efficacy of melittin in combination with oxacillin against methicillin-resistant Staphylococcus aureus (MRSA)
Journal: Microorganisms
Section: Antimicrobial Agents and Resistance
Cover Letter in Response to Reviewer Comments and Suggestions
Dear Guest Editors and Reviewer,
Thank you very much for reviewing our manuscript. On behalf of all the authors, we also greatly appreciate the reviewers for their complimentary comments and suggestions. Below are our point-by-point responses.
# Reviewer 1
The manuscript titled “Synergistic antibacterial efficacy of melittin in combination with oxacillin against methicillin-resistant Staphylococcus aureus (MRSA)” written by Pereira et al. focuses on the synergistic effect of the AMP melittin associated with oxacillin against MRSA. The topic of the antibiotic resistance is very current. In the last years many reaserches had the aim to develop new strategies able to overcome this global threat. The manuscript is well organized and the resullts reported could be of interest for the scientific community.
Only minor revisions are required before the publication in the journal:
- Introduction: line 44 needs a reference (e.g. International Journal of Molecular Sciences, 2023, 24(5), 4872). Some data on the known antibacterial and anti-biofilm properties of the AMP melittin should be reported (add the reference Nat Prod Res. 2022 Dec;36(24):6381-6388). Authors should report in the introduction Melittin’s antibacterial and antibiofilm activity alone and/or in combination with gentamicin, ciprofloxacin, rifampin, and vancomycin on biofilm-forming MDR-P. aeruginosa and MDR-MRSA strains ( Front. Microbiol. 14:1030401. doi: 10.3389/fmicb.2023.1030401). At line 53 add the recent reference on the antibiotic resistance of biofilm: Current Medicinal Chemistry, 2022, 29(25), pp. 4307–4310.
Response: We appreciate your contribution. The introduction has been rewritten, and the cited references have been included.
- Line 133: correct CO2
Response: The term “CO2” was corrected.
- Table 1: the concentration must be specified (µg/mL)
Response: Concentration information in µg/mL has been included.
- Figures 5 and 7 are not very clear, enhance their quality and the readability
Response: The Figure 5 was generated using the FACSCalibur BD® and its associated system software. The images were created within the BD® software, for which we no longer have access. However, a new image has been added. We have updated the captions for both figures to improve reader comprehension. All the figures included in the article are available in a high-quality zip folder, featuring a resolution of 300 dpi or greater, with pixel dimensions exceeding 1000.
- Line 403: correct “antibactericidal” in “bactericidal” or “antimicrobial”
Response: “Antibactericidal” was replaced by “bactericidal”.
- Ref 22: the year should be in bold
Response: We appreciate all your contributions to improve the article. The year was put in bold.

Reviewer 2 Report
Comments and Suggestions for Authors
In this paper, melittin and oxacillin were applied to methicillin-resistant Staphylococcus aureus, and their antibacterial activity, biosafety and potential antibacterial mechanism were discussed. However, the following issues need to be resolved.
1. It will be beneficial to provide 1-2 sentence background introduction of melittin and oxacillin.
2. In addition to antibiotics antimicrobial peptide conjugation, antibiotic peptide conjugation is another approach to enhance antibiotic activity, few very nice studies on the beta-lactam antibiotics conjugation with peptide should be discussed in the introduction, such as RSC Adv., 2012,2, 2480-2492, DOI: 10.1039/C2RA01351G; Peptide Science 2018 Vol. 110 Issue 3 Pages e24059, DOI: 10.1002/pep2.24059.
3. Lines 115 and 236 of the manuscript are incorrect in describing the detection wavelength of nucleic acid.
4. Apoptosis and necrotic cells can be observed in the flow chart (figure 5), which cannot prove that melittin and oxacillin are not cytotoxic when used alone or in combination.
5. It will be better to test the antibacterial stability in different media conditions, such as https://doi.org/10.1016/j.ejps.2020.105592
Comments on the Quality of English Languagesee comments
Author Response
Manuscript ID: microorganisms-2673353
Type of manuscript: Article
Title: Synergistic antibacterial efficacy of melittin in combination with oxacillin against methicillin-resistant Staphylococcus aureus (MRSA)
Journal: Microorganisms
Section: Antimicrobial Agents and Resistance
Cover Letter in Response to Reviewer Comments and Suggestions
Dear Guest Editors and Reviewer,
Thank you very much for reviewing our manuscript. On behalf of all the authors, we also greatly appreciate the reviewers for their complimentary comments and suggestions. Below are our point-by-point responses.
# Reviewer 2
In this paper, melittin and oxacillin were applied to methicillin-resistant Staphylococcus aureus, and their antibacterial activity, biosafety and potential antibacterial mechanism were discussed. However, the following issues need to be resolved.
- It will be beneficial to provide 1-2 sentence background introduction of melittin and oxacillin.
Response: Thanks for the suggestion. The introduction has been revised, integrating additional details about melittin and oxacillin.
- In addition to antibiotics antimicrobial peptide conjugation, antibiotic peptide conjugation is another approach to enhance antibiotic activity, few very nice studies on the beta-lactam antibiotics conjugation with peptide should be discussed in the introduction, such as RSC Adv., 2012,2, 2480-2492, DOI: 10.1039/C2RA01351G; Peptide Science 2018 Vol. 110 Issue 3 Pages e24059, DOI: 10.1002/pep2.24059
Response: Thanks for your contribution. The introduction was rewritten, and the recommended references have been included.
- Lines 115 and 236 of the manuscript are incorrect in describing the detection wavelength of nucleic acid.
Response: The correct optical density is 260 nm, and it was modified in the methodology.
- Apoptosis and necrotic cells can be observed in the flow chart (figure 5), which cannot prove that melittin and oxacillin are not cytotoxic when used alone or in combination.
Response: The flow chart depicted the apoptosis and necrotic cells following treatment with melittin, oxacillin and melittin associated with oxacillin (mel+oxa), the percentages of the apoptotic and/or necrotic cells were calculated, ranging from 1.06% to 6.11%, which were not statistically significant. This indicates that at their minimum inhibitory concentration (MIC) and 2 times the MIC value, the tested products did not induce significant apoptosis and necrosis in HaCaT keratinocyte cells. The conclusion and Figure 5 legend was revised to emphasize that the products did not show apoptosis or necrosis in HaCaT cells instead did not show cytotoxicity.
- It will be better to test the antibacterial stability in different media conditions, such as https://doi.org/10.1016/j.ejps.2020.105592
Response: We appreciate the suggestion. However, conducting an antibacterial stability test of melittin in different media conditions was not among our objectives. Our group is currently planning another article that will focus on describing the structure, stability in different media conditions, and conjugation of melittin with nanostructures.

Reviewer 3 Report
Comments and Suggestions for Authors
The authors performed good work with interesting results; however, I have some concerns and comments that need consideration.
1. The introduction section needs more refinement. Please provide more information about S. aureus and its resistant form, MRSA, for example, bacterial type, etc. Also, please provide information about the current problem and the reason for antibacterial resistance to standard antibiotics, which in turn led to the need for finding new sources such as natural antibacterials with less resistance. I recommend the authors use the reference (DOI: 10.1016/j.sajb.2016.10.001) to extract the required information.
2. The authors used two bacterial strains: one standard ATCC MRSA and one isolated MRSA. Do you think it is enough to perform a study on one isolate without performing bacterial identification studies? please explain.
3. In the materials and methods section, the authors did not show how they performed the antibacterial combinatory effects. please clarify.
4. Also, in the material and methods section, the authors did not provide how they performed protein-protein interaction networks, where their results are shown in Figure 8. Please clarify.
5. In a supplementary file, please provide the data (chromatographic data) obtained from the mass spectrometry analysis.
6. The work has lots of results; thus, I recommend the authors highlight the effective concentrations of the tested compound, which can be used by other researchers to build upon them in further studies. This can be highlighted in the conclusion section.
Comments on the Quality of English LanguageI recommend the authors double-check the full text for grammatical and typing errors and seek the help of a native English speaker.
Author Response
Manuscript ID: microorganisms-2673353
Type of manuscript: Article
Title: Synergistic antibacterial efficacy of melittin in combination with oxacillin against methicillin-resistant Staphylococcus aureus (MRSA)
Journal: Microorganisms
Section: Antimicrobial Agents and Resistance
Cover Letter in Response to Reviewer Comments and Suggestions
Dear Guest Editors and Reviewer,
Thank you very much for reviewing our manuscript. On behalf of all the authors, we also greatly appreciate the reviewers for their complimentary comments and suggestions. Below are our point-by-point responses.
# Reviewer 3
- The introduction section needs more refinement. Please provide more information about S. aureus and its resistant form, MRSA, for example, bacterial type, etc. Also, please provide information about the current problem and the reason for antibacterial resistance to standard antibiotics, which in turn led to the need for finding new sources such as natural antibacterials with less resistance. I recommend the authors use the reference (DOI: 10.1016/j.sajb.2016.10.001) to extract the required information.
Response: Thank you for your suggestions. We have taken them into account and subsequently revised the introduction. Additionally, we have incorporated the recommended reference into our work.
- The authors used two bacterial strains: one standard ATCC MRSA and one isolated MRSA. Do you think it is enough to perform a study on one isolate without performing bacterial identification studies? please explain.
Response: The MRSA clinical isolate was previously characterized by the detection of the mecA gene using PCR (Figure 1) and has been stored at the Department of Chemical and Biological Sciences, specifically in the Microbiology and Immunology sector at the Institute of Biosciences of Botucatu (IBB), São Paulo State University (UNESP).

Figure 1. PCR for detection of gene mecA. 1 – positive control, 2 – negative control, 3 – clinical isolate of MRSA (positive for mecA).
- In the materials and methods section, the authors did not show how they performed the antibacterial combinatory effects. please clarify.
Response: For the antibacterial combinatory effects, we used 25% of the MIC of melittin combined with 25% MIC of the oxacillin and 25% of the MIC of melittin combined with 25% MIC of the cephalothin. The material and methods section “2.3. Bacterial Growth Curve Assay and Time-Kill Assay” was revised to improve comprehension.
- Also, in the material and methods section, the authors did not provide how they performed protein-protein interaction networks, where their results are shown in Figure 8. Please clarify.
Response: The software STRING was used to construct protein-protein interaction networks, and the "2.13. Analysis of Mass Spectrometry Data" section in the materials and methods was revised.
- In a supplementary file, please provide the data (chromatographic data) obtained from the mass spectrometry analysis.
Response: All mass spectrometry data from the Computer Science and Engineering department at the University of California, San Diego, have been uploaded to the MassIVE Repository. You can access the data through the following link: (https://massive.ucsd.edu/ProteoSAFe/dataset.jsp?task=47d4f410165d4ab19b872bf391928182). We have now made the data publicly accessible. The dataset AF1 represents the control group with only MRSA, while the remaining groups involve MRSA with specific treatments: AF2 (oxacillin), AF3 (melittin), and AF4 (melittin combined with oxacillin). This information has been included in the article.
- The work has lots of results; thus, I recommend the authors highlight the effective concentrations of the tested compound, which can be used by other researchers to build upon them in further studies. This can be highlighted in the conclusion section.
Response: We appreciate all the suggested contributions aimed at enhancing our article. The concentrations tested at MIC and 2x values of the combination of melittin with oxacillin (mel+oxa) were highlighted in the conclusion section.

Round 2
Reviewer 3 Report
Comments and Suggestions for Authors
The manuscript has been significantly improved.
Comments on the Quality of English LanguageThe English usage is fine, however, minor refinement can be done during the proofreading stage.